# Immune Microenvironment and Immunotherapeutic Management in Virus-Associated Digestive System Tumors

**DOI:** 10.3390/ijms232113612

**Published:** 2022-11-06

**Authors:** Panagiotis Sarantis, Eleni-Myrto Trifylli, Evangelos Koustas, Kostas A. Papavassiliou, Michalis V. Karamouzis, Athanasios G. Papavassiliou

**Affiliations:** 1Department of Biological Chemistry, Medical School, National and Kapodistrian University of Athens, 11527 Athens, Greece; 2First Department of Internal Medicine, 417 Army Share Fund Hospital, 11521 Athens, Greece

**Keywords:** tumor immune microenvironment, HBV, HCV, EBV, HPV, CMV, JCV, digestive system, immunotherapy

## Abstract

The development of cancer is a multifactorial phenomenon, while it constitutes a major global health problem. Viruses are an important factor that is involved in tumorigenesis and is associated with 12.1% of all cancer cases. Major examples of oncogenic viruses which are closely associated with the digestive system are HBV, HCV, EBV, HPV, JCV, and CMV. EBV, HPV, JCV, and CMV directly cause oncogenesis by expressing oncogenic proteins that are encoded in their genome. In contrast, HBV and HCV are correlated indirectly with carcinogenesis by causing chronic inflammation in the infected organs. In addition, the tumor microenvironment contains various immune cells, endothelial cells, and fibroblasts, as well as several growth factors, cytokines, and other tumor-secreted molecules that play a key role in tumor growth, progression, and migration, while they are closely interrelated with the virus. The presence of T-regulatory and B-regulatory cells in the tumor microenvironment plays an important role in the anti-tumor immune reaction. The tumor immune microenvironments differ in each type of cancer and depend on viral infection. The alterations in the immune microenvironment caused by viruses are also reflected in the effectiveness of immunotherapy. The present review aims at shedding light on the association between viruses and digestive system malignancies, the characteristics of the tumor immune microenvironment that develop, and the possible treatments that can be administered.

## 1. Introduction

Cancer constitutes a major public health problem worldwide and it is estimated that almost 20 million new cancer cases and nearly 10 million cancer-related deaths occurred in 2020 [1]. Infectious factors are involved in 17.8% of all cancers (viruses for 12.1%, bacteria for 5.6%, and helminths for 0.1%) [2,3]. The distribution of viruses could be in several organs/tissues, while in the case of virus-related cancer the tumor microenvironment (TME) could be significantly altered, depending on the type of virus. The human digestive system bears an enormous number of different viral particles (>1015) [4]. Many DNA viruses (e.g., Herpesviridae family) can induce persistent infections, whilst anelloviruses have not been linked with a specific pathology [5]. On the other hand, RNA viruses are more likely to cause acute infections [6]. There are several viruses that have a great importance, due to the fact that they can establish a long-term correlation with their host and can induce an inflammatory status [7]. Indeed, chronic inflammation can stimulate a carcinogenic background, which can lead to tumorigenesis [8]. It must be emphasized that viruses that integrate their genetic material into the human DNA can cause dysregulation of oncogenes and/or inactivation of tumor suppressive genes (direct carcinogenesis) [9].

Seven viruses have been significantly associated with carcinogenesis, including hepatitis B and C viruses (HBV and HCV), human papilloma virus (HPV, with types 16, 18, 33, and 45 as the most dominant), Epstein-Barr virus (HHV4/EBV), human T-cell lymphotropic virus type I (HTLV-I), human herpesvirus 8, also known as Kaposi’s sarcoma virus (HHV8), and Merkel cell polyomavirus (MCV) [4,10]. Viruses which are associated with the digestive system are HBV, HCV, EBV, HPV, John Cunningham virus (JCV), and cytomegalovirus (CMV) [6,11,12]. For example, HPV and EBV directly cause oncogenesis by expressing oncogenic proteins encoded in their genome. On the contrary, HBV and HCV are correlated with indirect carcinogenesis by causing chronic inflammation of the infected organs [13] (Figure 1).

TME contains various immune cells, endothelial cells, and fibroblasts, as well as several growth factors, cytokines, and other tumor-secreted molecules. Some of the cells that are located in the tumor surrounding stroma are myeloid-derived suppressor cells (MDSCs), tumor-infiltrating lymphocytes (TILs), tumor-associated macrophages (TAMs), cancer-associated fibroblasts (CAFs), as well as many molecules that are secreted from the malignant cells [14]. Additionally, the presence of T-regulatory (Treg) and B-regulatory (Breg) cells in the TME constitutes an impediment to the physiological anti-tumor immune reaction phenomenon, which is mainly attributed to the deregulation or inhibition of T-effector action by the expression of FOXP3 on Tregs, as well as to interleukin-10 (IL10) secretion by Bregs that suppresses the cytotoxic effect of T-cells. All the aforementioned cells present possible therapeutic targets for anti-neoplastic agents, as they promote tumor development, migration, and angiogenesis [15]. A better understanding of the TME components and the mechanisms that are implicated in tumor escape is considered pivotal for the management of HCC. The interrelation of TME with immune responses has a pertinent role in disease progression, while it contains quite a few targets for anti-cancer therapy. The immune response is a complex process that is compounded by several steps, including: (i) The asymptomatic step, (ii) the balance step, and (iii) the tumor escape. In the first step, immune cells attempt to recognize and eliminate the cancer cells, which can be achieved initially via CD8+ and CD4+ T-cells, NK cells, as well as T-helper 1 cells, and subsequently via the formation of antibodies against the antigens on the surface of cancer cells. However, if the above cells cannot recognize and eliminate the malignant cells, there is the following step of balance, whereby there is tumor growth and progression by avoiding the immunosurveillance mechanisms. This phenomenon is mainly attributed to the expression of immunogens on the cancer cell surface. The next step includes tumor escape, in which cancer cells continue to grow, regardless of the administration of immunotherapeutic agents. The evasion of immunosurveillance is achieved via the expression of inhibitory checkpoints on the cancer cells, that recruits Tregs, Bregs, TAMs, and MDSCs [16].

The human virome is a crucial part of the human microbiota and develops as a new and important field of study. Herein, we provide an updated overview of the digestive system malignancies that are related to viruses, their impact on TME that promotes carcinogenesis, and the opportunities for immunotherapeutic management.

## 2. Hepatitis B Virus (HBV)

Hepatitis B virus infection constitutes one of the most severe and predominant infectious diseases as well as one of the main etiologies of chronic hepatitis infection (CHI). Additionally, it is widely manifested in the general population, exceeding 2 billion individuals globally, with approximately 360 million inactive HBsAg carriers, while it is considered the second most prevalent cause of death due to malignancy, and the fifth most commonly diagnosed cancer [17]. The majority of hepatocellular carcinoma (HCC) cases have an etiopathogenic background of HBV, especially in the population of Africa and China, whilst less than 10% of total cases are non-HBV-related [18]. The viral genome is a double-stranded partially circular DNA that encodes several proteins, such as the minimum X protein, HBeAg, and HBsAg proteins [19]. Notably, HCC can be developed in cirrhotic and non-cirrhotic patients, in which hepatocarcinogenesis occurs directly by the persistence of viral infection.

### 2.1. HBV-Associated Hepatocellular Carcinoma (HCC)

Hepatocarcinogenesis that is associated with HBV infection is mainly attributed to the persistent inflammation of hepatic parenchyma and the subsequent fibrotic injury, which leads to cirrhosis, in a background of deregulation of multiple signaling pathways, modified immune cell functions, the influence of TME, as well as the long-term oxidative stress, resulting from the persistence of the viral infection. Cirrhosis is considered the major predictive factor for HCC development, implying a gender disparity and the age of the patient [20,21].

Moreover, a better knowledge of the immunoregulatory checkpoints that are presented, which significantly modify the anti-neoplastic immune response, is crucial for the management of this malignancy. In particular, PD-1, CTLA-4, as well as lymphocyte activation gene 3 (LAG-3), B7-H3/H4, and A2AR constitute some of the inhibitory checkpoints, in comparison with the stimulatory that are CD137, CD27, CD122, as well as CD28 and CD40. However, all the above can be expropriated by cancer cells that escape immunosurveillance [22].

It has been demonstrated that chronic infection with HBV noticeably modifies the activity of TME cells, such as TILs, while elevated levels of T-regulatory cells significantly alter the activity of cytotoxic T-lymphocytes (CTLs), which leads to immunosuppression that further promotes tumor growth and progression [23]. Moreover, chronic immunosuppression that is induced by HBV infection is illustrated by the impairment of NK cells [24]. This suppression and modification of the functional state of NK cells are attributed to hepatic MDSCs, which interact with NKp30 on NK cells. Furthermore, MDSCs enhance the amount of Tregs, which modify the CTLs function, while they promote the exhaustion of CD8+ T-cells, via expressing TIM-3, CTLA-4, and PD-1, as well as suppressing CD4+ T-lymphocytes [23].

Additionally, there is a high expression of immune checkpoints, such as cytotoxic T-lymphocyte antigen 4 (CTLA-4) on Tregs, which modify the function of antigen-presenting cells (APCs), via IL10 and TGF-b, which cannot stimulate CTLs. Based on a study that compared negative with positive pre-S2 patients, the latter type of tumors had a similar amount of CTLs to the former type; however, it presented an increased amount of forkhead box P3 (Foxp3) and CD25+ and CD4+ cells, as well as a decreased number of cells that were expressing granzyme B. Moreover, it was shown that the positive-pre-S2 mutant HCC cases have a dismal prognosis, in comparison with negative-pre-S2 cases. Of note, Tregs regulation is closely associated with transforming growth factor-β1 (TGF-β1), which further enhances the immunosuppressive effect of Tregs in TME via promoting their recruitment and differentiation. Immunosuppression in the TME is further aggravated via the direct effect of TGF-β1 on cytotoxic T-lymphocytes, with TGF-β1 secretion being promoted by pre-S2 mutant HCC [23,25]. Furthermore, it was reported that the HCC microenvironment is considered significantly more immunosuppressive in cases where HBV coexists, in comparison with HCC cases that are not related to viral hepatitis [26].

Noteworthy, HBV-HCC tumor analysis does not show any aberration in T-cell immune response pathways, in comparison with HCV-HCC tumors; however, complement pathway and B-cell activation pathways were found to be downregulated [27]. Additionally, in HBV-HCC tumors, some genes that are associated with macrophage/monocyte activation pathways were found to be upregulated, such as S100A11, MIF, S100A10, as well as S100A4, while some other genes were also upregulated, including TALDO1, MIF, S100A11, NOS1, and S100A6, that are involved in natural immunity [27,28].

The aforementioned characteristics open new therapeutic opportunities for the management of HBV-related HCC cases, such as by manipulating the expression of PD-1 on Tregs, via immunotherapeutic agents, including immune checkpoint inhibitors, such as anti-PD-1 inhibitors. A CTLA-4 inhibitor known as tremelimumab (phase II clinical trial, NCT01008358) has been studied for advanced HCV-related HCC, which is a well-tolerated agent that provides a 76.4% disease control rate (DCR), decreases the viral load, as well as provides 8.2 months of overall survival (OS) [29,30].

Pembrolizumab (a PD-1 inhibitor) constitutes a second-line treatment for advanced HCC, which has been studied in KEYNOTE 224 phase II study, also for HBV- and HCV-related HCC, which provided 12.9 months of OS and did not induce reactivation of the infections [31]. It has been shown that HBV-infected patients with HCC had an increased OS in comparison with patients who received a placebo, an effect that was not observed in HCV-infected patients with HCC. Another combination is IBI305 (biosimilar of bevacizumab) and sintilimab, and another PD-1 inhibitor in phase II/III (NCT03794440) in comparison with sorafenib is a first-line choice for HBV-related unresectable HCC cases [32]. Additionally, another modality is cytokine-induced killer cells (CIKs) infusion (phase III trials NCT01749865, NCT00769106), which is a cell-based therapy [33,34]. Another cell-based therapy that was tested is the utilization of TCR-engineered cells (TCR redirected therapy) against HBV antigens, such as HBV x protein, which show favorable results and constitute a promising therapeutic strategy [35,36,37]. Finally, oncolytic therapy is another immunotherapeutic modality that can be utilized in HBV-related or HCV-related HCC, such as JX-594, which showed favorable effects in the infected patients and a significant improvement in OS, especially when the dose of JX-594 was increased [38,39,40,41].

### 2.2. HBV-Associated Cholangiocarcinoma (CCA)

Cholangiocarcinoma constitutes a group of malignancies including extrahepatic and intrahepatic distinct entities. Intrahepatic cholangiocarcinoma (iCCA) is not the most frequent form of cholangiocarcinoma (10–20%), while it is frequently associated with prior HBV infection. Chronic inflammatory state is considered crucial for iCCA development regardless of the etiopathogenic mechanism that induces it. This state favors carcinogenesis via promoting genetic and epigenetic aberrations [42,43]. Carcinogenesis is facilitated by the overregulation of several growth factors and cytokines that promote increased cell multiplication, as well as angiogenesis. Some of the mediators that are recognized include tumor necrosis factor (TNF), vascular endothelial growth factor (VEGF), interleukin-6 (IL6), which has a key role as a growth factor for tumor cells, hepatocyte growth factor (HGF), and transforming growth factor beta (TGF-β) [44,45]. Of note, the expression of PD-L1 opens opportunities for anti-neoplastic management with immunotherapeutic agents [46]. It has been reported that chronic HBV infection predisposes iCCA development, especially in young-aged males, while this HBV-related iCCA is usually characterized by a favorable prognosis. Some major mutant genes are TP53 and KRAS, as well as MYC and IDH1 and IDH2 genes. In general, chronic biliary inflammatory state predisposes several mutations except the aforementioned, such as ARID1A, ROBO2, as well as MLL3, RNF34, FGFR fusions, and CDKN2A mutation. HBV infection is closely associated with TP53 mutation, while the cases with TP53 and KRAS mutant genes demonstrate a dismal prognosis [27,47].

Moreover, TME has a chief role in CCA, which promotes the recruitment and activation of TAMs that influence several cells, including stromal cells and malignant cholangiocytes, and are closely associated with neoangiogenesis and metastatic dissemination. Μoreover, TAMs influx is closely associated with worrisome prognosis, increased aggressivity, and recurrence rates. TAMs could serve as a potential therapeutic target for the management of disease progression [48]. Furthermore, hyperactivation of PD1/PD-L1 in iCCA tumors is a negative prognostic biomarker. Elevated neutrophils and iDC predicted poor survival, while the increased number of lymphocytes (CD4+ T-cells, CD8+ T-effector cells, and B-cells) were related to good prognosis. Finally, ICC patients with HBV infection have a more positive prognosis compared with ICC patients without HBV infection. Song et al. identified XCL1 + CD8+ T-cells that are linked with better patient survival. From the above, PD1/PD-L1-targeted immunotherapy is an encouraging treatment option for ICC patients [49,50,51,52].

### 2.3. HΒV-Associated Colorectal Cancer (CRC)

Colorectal cancer has been shown to be closely associated with viral hepatitis B, which significantly increases the risk for colorectal carcinogenesis, especially in the Chinese population [53]; therefore, these patients are in need of frequent screening. A higher risk of CRC development was also shown in studies performed in the Korean population [54,55]. Based on large cohort studies, an increased risk for developing CRC in patients with HBV infection was reported, implying the impact of HBV virus in colorectal carcinogenesis; however, with a background of modified risk factors, such as tobacco and alcohol consumption, age, gender, as well as body mass index (BMI). In particular, CRC risk was not significantly associated with female gender, young-aged or elderly, non-consumers of tobacco, alcohol consumption, and increased BMI (25–29.9 and 30–34.9). In HBV seropositive patients in Taiwan, it was shown that they had a higher risk (36% higher) of developing CRC, in comparison with seronegative (HBsAg negative) patients [53]. Moreover, the risk of synchronous colorectal liver metastasis (CRLM) was enhanced in cases with chronic HBV infection, demonstrating a dismal prognosis, in comparison with patients with occult hepatitis B infection (OBI) that exhibited a lower risk of metachronous CRLM and a longer free of disease post-surgical survival. These reports imply that active HBV replication enhances the risk of developing synchronous CRLM, due to the fact that during the migration of colorectal malignant cells into the circulation, the hepatic immune responses are intensified due to HBV infection, resulting in an increased risk in developing secondary liver cancer, and in other circumstances, primary liver cancer [56]. Another pathogenetic mechanism that promotes CRC metastatic dissemination is the overregulation of immune checkpoint PD-L1 and PD-1, resulting in the tumor immune escape phenomenon [56]. A study by Cheng et al. revealed that efficacy and OS were similar between patients with HBV infection and non-HBV patients receiving anti-PD-1 therapy. Therefore, CRC-HBV-positive patients can benefit from immunotherapy. Nevertheless, further research is needed on the correlation between HBV infection, immune microenvironment, and CRC [57].

### 2.4. HBV-Associated Pancreatic Adenocarcinoma (PAC)

Pancreatic adenocarcinoma (PAC) constitutes a quite lethal malignancy, the third most frequent causal agent for malignancy-related deaths in the United States in recent years, demonstrating a worrisome prognosis and a low 5-year survival rate of less than 10% [58]. Despite the most frequent risk factors for pancreatic carcinogeneses, such as chronic alcohol consumption, diabetes mellitus, as well as aging and obesity, HBV constitutes another risk factor for PAC development. The interplay between HBV infection and PAC was considered controversial for many decades; however, based on meta-analytic studies, it has been proved that the risk of PAC is increased in patients with HBV infection [59,60,61]. The pathogenetic mechanism between HBV infection and PAC is not yet specified. The transport of the virus from hepatic parenchyma to the pancreatic anatomical region is mainly attributed to the commonly shared ductal and vasculature system, with HBV being transferred to pancreatic parenchyma, which leads to a chronic inflammatory state that promotes pancreatic carcinogenesis. Another hypothesis is the common embryonic origin of hepatocytes and pancreatic cells, as well as the molecular background of HBV infection, which induces various genetic and epigenetic aberrations [60,62,63]. Unfortunately, there are no studies focusing on HBV-associated PAC and the immune microenvironment.

## 3. Hepatitis C Virus (HCV)

Hepatitis C virus, a member of the genus Hepacivirus within the Flaviviridae family, constitutes a single-stranded RNA virus, which is significantly hepatotropic, with a variety of six genotypes, which are closely associated with the duration of the HCV-induced HCC development. Specifically, an increased risk for hepatic carcinogenesis is closely associated with genotypes 1 and 3, with an 80% increase in the risk, as well as genotype 6 which is mostly reported in developing countries [64,65]. Approximately 80% of patients infected with HCV develop a chronic infection, while 15% develop acute infection [66]. Of these patients who develop chronic infection, 5–20% will develop cirrhosis in the next 5 to 20 years, and 1–2% will develop HCC per year [67].

### 3.1. HCV-Related HCC

HCV is considered another major factor that leads to liver cirrhosis and eventually HCC, with increased morbidity and mortality rates. An increased risk of HCC development is closely associated with occult HBV infection on a background of chronic HCV infection, as well as coinfection with human immunodeficiency virus (HIV), while patients with only HCV infection have a 17-fold higher risk for HCC, in comparison with seronegative patients [27,68,69]. Moreover, HCV patients with an active state of HBV replication, have a 2-fold increased risk for HCC and an increased mortality rate, compared with patients with no active HBV replication. Whereas, HIV coinfected HCV patients have a more rapid evolution to HCC, which is directly proportional to CD4 count [70,71,72]. Similarly, patients with a history of alcohol abuse had an increased risk of HCV-related hepatocarcinogenesis, especially for HCV patients with DB and obesity with adiponectin resistance, the risk for HCC is two or three times higher or up to four times higher, respectively [73,74].

On a molecular basis, the HCV oncogenic effect on liver parenchyma is attributed to HCV-related proteins, such as the non-structural proteins and the core protein, which modify the host genome and alter various signaling, immune, and cell-cycle pathways, while HCV via secreting pro-inflammatory cytokines, such as interferon and TGF-β promotes disease progression and hepatic carcinogenesis. Specifically, the former proteins are associated with the activation of hepatic stellate cells (HSCs), via TGF-β, which induces fibrinogenesis and enhancement of the inflammatory state [75,76,77]. Moreover, the latter protein induces the suppression of the TP53 tumor suppressor gene, as well as Retinoblastoma gene mutation significantly induces hepatic cancer cell development and growth, which is also attributed to the modified cell cycle that promotes the accumulation of several mutations. The expression of cancer checkpoints, several mutations, such as in β-catenin, Retinoblastoma, and TP53 genes, as well as the mutation of the telomerase gene constitute some of the promoters of HCC [78,79,80]. Table 1 presents an overview of HCV proteins that induce several deregulations that lead to hepatic carcinogenesis.

Furthermore, based on the analysis of HCV tumors, the expression of 2481 genes was altered, while various signaling pathways (up to 14) which are closely associated with adaptive and innate immunity were found to be modified, with 95% of the genes exhibiting downregulation. Some of these altered immune pathways include Th1/2 activation pathways, the maturation pathway of dendritic cells, as well as the signaling pathway of primary immunodeficiency. LXR/RXR pathway, as it was previously referred to, is modified in both types of viral hepatitis-related tumors, which is crucial for the metabolism of lipids, as well as the pathway of acute phase response, the signaling pathway of primary immunodeficiency, as well as the activation pathway of HSCs [27,81]. Additionally, CD86, CD274, as well as CD3E are some of the genes that are found to be downregulated in HCV-HCC tumors, which have a key role in the T-cell activation pathway. Similarly, HAVCR2, CD53, GATA3, CD96, as well as TAGAP, CD7, VSIG4, CD8B, and VTCN1, were found to be downregulated in these tumors, which also have a significant role in T-cell function [27,82,83]. Some other genes that are associated with T-cell differentiation and cytokine and chemokine secretions are IL18R1, TGFβ3, IL20RA, CCL14, IL2RA/B, CXCL14, TGFA, as well as IL7R, IL15RA, IL21RA, and IL17RE. Similarly, IL1RAP, IL4R, IL18RAP, as well as IL1RL1, IL27, and IL21EA, were also identified [27,75,84,85].

Nonetheless, in HCV-related HCC there is a noteworthy deregulation of T-cell response and activation pathways, resulting in the suppression of T-cell-associated genes, compared with HBV-related tumors, which did not present any of these aberrations [86]. Similarly, with HBV, HCV also alters the HCC TME via T-cell impairment, with CD8 and CD4+ T-lymphocytes presenting dysfunction. HCV infection induces TGF-β and PD-L1/PD-1 overregulation and lessens T-cell responses that promote immunosuppression in HCC. The secretion of TGF-β has a negative impact on T-effector cells, while the PD-1/PD-L1 axis is closely related to cancer immune escape and is also associated with the severity of the malignancy and the poor outcome [87,88].

Finally, camrelizumab is a PD-1 inhibitor that is currently assessed in a phase II trial (NCT02989922) [89,90], as well as tislelizumab (NCT02412773) in a phase III trial [90]. The former HCV patients with HCC exhibited a good response, with a median OS of 14.4 months [91]. Another modality is the utilization of vaccine-based treatment for the enhancement of T-cell response, which is impaired in cases of infected HCC patients, such as AFP-based HCC vaccine, which was tested on HBV- and HCV-related HCC patients, as well as a dendritic cell-based vaccine (DC vaccine) as an adjuvant, which is studied in a phase II trial for patients with resected HCC [92]. Glypican-3 (GPC3) is another vaccine that is assessed in phase 1 trial in patients with a history of HCV and HBV, which induces T-cell response against the glypican-3 on cancer cells [93,94].

### 3.2. HCV-Related Cholangiocarcinoma (CCA)

There are several studies that demonstrate the interplay between HCV and CCA, including the intra- and extra-hepatic types [95,96]. The oncogenic properties of HCV proteins significantly modify the functions of cholangiocytes, promoting their transformation into malignant cells. There is a hypothesis that the cholangiocytes are more vulnerable to HCV infection, which is possibly attributed to specific receptors on cholangiocytes that permit the adhesion and the invasion of the virions. Another hypothesis includes the direct effect of HCV proteins on cholangiocytes, leading to a chronic inflammatory state, as well as fibrotic injury, which promotes cholangiocarcinogenesis. Epithelial-mesenchymal transition (EMT) constitutes a key mechanism that promotes CCA development, in which there are phenotypic and functional alterations and transformation of epithelial cells into mesenchymal. It was demonstrated in QBC939 CCA lines that HCV core protein is closely associated with the phenomenon, while there is an aberrant expression of fibronectin, E-cadherin, as well as vimentin, which are related to cancer cell migratory behavior [97,98,99]. Regarding HCV-related cholangiocarcinoma and immune microenvironment, there are no specific studies. As mentioned above, immunotherapy helps in HCV-related HCC and there are similarities between HBV and HCV. Consequently, there is the possibility of similarities in the immune microenvironment caused by HCV in CCA [50,52].

### 3.3. HCV-Related Pancreatic and CRC Cancers

It has been shown that HCV infection, as well as HCV and HIV coinfection, are considered risk factors for developing several digestive system cancers including CRC and PAC. Based on several cohorts and case-control studies, it has been reported that HCV patients have a significantly increased risk of developing rectal cancer and an elevated risk of endoscopic diagnosis of CRC. However, this issue is controversial as other studies do not demonstrate the above phenomena. Similarly, the risk of developing PAC is also increased for HCV patients, especially if they have a history of tobacco consumption [61]. In both of the above digestive system cancers, even after seroclearance, the risk of carcinogenesis remains high. From a molecular point of view, the suppression of the Tp53 gene is identified, which is mainly attributed to the HCV core protein, which subsequently leads to genetic aberrations and carcinogenesis [100,101]. As in the cases of HBV-related pancreatic cancer and HBV-related CRC, there are no studies on the immune microenvironment.

## 4. John Cunningham Virus (JCV) and Human Cytomegalovirus (CMV)-Associated CRC

The John Cunningham virus or human polyomavirus 2 has been mainly identified in the gastrointestinal tract, where it can remain latent. It encodes three capsid VP1–3 and two T-antigen proteins, as well as agnoprotein. It has been shown that JCV is closely related to many malignancies, including CRC, via encoding the T-antigen (Ag) gene [102]. The expression of T-Ag is implicated in chromosomal instability, which leads to carcinogenesis and the downregulation or silencing of tumor suppressor genes, such as p53 and APC, as well as the pRb family [103]. The encoding proteins of the T-Ag gene have a crucial regulatory role in cell multiplication, as well as in viral dissemination. It is hypothesized that JCV is reactivated in patients with latent digestive system infection, which subsequently leads to CRC [103,104]. There is a prominent immune response against T-Ag proteins, including the stimulation of T-cytotoxic and helper cells. However, in patients with no malignancy, the stimulation of Th1 cells is higher [105]. Finally, based on a meta-analysis of case-control studies of JCV infection and CRC risk, this viral infection, presenting T-Ag protein expression elevates 10-fold the risk of CRC development, implying the necessity of further studies for the development of novel therapeutic strategies and the limitation of this phenomenon [106].

Additionally, human cytomegalovirus constitutes a herpesvirus, which is characterized by persistent latent infection and is implicated in the aggregation of antigen-related T-cell pools, the so-called T-cell memory inflation. The emerging role of CMV in various cancers including CRC is in the spotlight, while CMV is presented in 40% of the total CRC bioptic specimens [107,108]. CMV proteins have been demonstrated in bioptic specimens of brain secondary lesions in CRC patients, in which the virus might be reactivated during the administration of chemotherapeutic agents, which might require treatment [109]. It has been reported that cancer patients with various types of malignancies had an aberrant amount of CMV-related CD8+ T-cells, especially in cases with more end-stage disease [108,110]. CMV reactivation might occur later in adult life with a concomitant increase in memory CD8+ T-cells, the so-called memory inflation against the persistent CMV-DNA load in the host. The initial viral dose is closely related to the T-cell memory inflation, with lower levels of viral load being proportionally associated with latent viral reservoirs, as well as with T-cell multiplication [111]. Furthermore, the presence of CMV in malignant tissue is closely related to worrisome prognosis, while the genetic material of this virus is profoundly found in the malignant epithelium in comparison with non-neoplastic. Disease-free survival period is considered higher in patients under 65 years old compared with older patients [112,113]. A more aberrant inflammatory response within the tumor was demonstrated in CMV-positive cases with upregulated IL1 levels, enhancing tumor proliferation, neoangiogenesis, and metastatic dissemination. Particularly, CMV-infected cells can escape the immunosurveillance via encoding various proteins, such as US11, US2, US6, and US3, which are implicated in the downregulation of MHC I molecules on cells, which are infected, while except for IL1, they also upregulate the expression of IL17 and modify the anti-neoplastic immune response in the TME. Meanwhile, T-cell-mediated immunity is altered via the viral immunomodulatory effect, which is also implied by the fact that the expression of MMPs that promotes invasion and metastatic dissemination is modified, as well as encoding several other proteins that inhibit apoptosis, such as UL37-UL38, while a higher MMP-1 level is found in CMV-positive specimens. A better understanding of the CMV immunomodulatory role in CRC opens new therapeutic strategies via targeting this potentially oncogenic virus [113]. Nevertheless, further research is needed for the aforementioned viruses and their implication in TME.

## 5. Epstein-Barr Virus (EBV)

The Epstein-Barr virus (EBV, human gammaherpesvirus 4) is one of the nine known human herpesvirus types of the Herpes family [114]. EBV is a double-stranded DNA virus affecting more than 90% of adults and is responsible for 1.5% of total cancers. Latent EBV infection is recognized to be related to epithelial cancers and multiple lymphoid malignancies. Common EBV-associated epithelial cancers include EBV-associated gastric cancers (EBVaGC), nasopharyngeal carcinoma (NPC), and lymphoepithelioma-like carcinoma (LELC). Rarely, EBV can cause breast cancer, salivary gland cancers, thyroid cancer, and hepatobiliary cancers [115]. Tumors have been classified based on the presence of TILs and PD-L1 expression into four different tumor immune microenvironments (TIMEs). They are type I (adaptive immune resistance), type II (immunological ignorance), type III (intrinsic induction), and type IV (tolerance) [116].

### 5.1. EBV-Associated Gastric Cancer (GC)

EBV is detected in approximately 10% of gastric cancer (GC) worldwide. The EBV genome exists in all cancer cells. Incidence of EBVaGC ranges depending on the region, (16–18% in the USA and Germany, and 4.3% in China). EBVaGC has discrete clinicopathological features; it appears in a higher percentage of men (71%) and has a generally diffuse histological type. A current study demonstrates a link between gastric cancer and EBV infection, while EBV-positive and EBV-negative tumors are closely related to tobacco abuse [117,118].

EBVaGC has a certain histological association with gastric carcinomas with lymphoid stroma (GCLS) [119,120]. GCLS has an intense and diffuse lymphocyte infiltration and is a poorly differentiated adenocarcinoma analogous to EBV-associated nasopharyngeal lymphoepithelioma. In a recent study, 124 tumors were tested by EBER1-ISH and from the 124 tumors, 12 (9.7%) were identified as EBVaGC [121]. Moreover, Zhao et al. have repeatedly identified the presence of EBV and H. pylori. in the mucosa of patients with moderate chronic atrophic gastritis [122].

EBVaGC is regularly removed by surgical resection since it is an undifferentiated-type cancer. Early EBVaGC has a low risk of lymph node metastasis [123]. A study presented that EBV-associated GC has a significantly low incidence of lymph node metastasis and a better prognosis compared with EBV-negative GC. An unadjusted Cox regression analysis shows that the median survival duration of EBV-positive GC patients is 8.5 years and for EBV-negative patients is 5.3 years [124].

Statistical analysis shows that EBVaGC with a small mutation burden is a subset of MSS GC and has good results to immune checkpoint therapy. Gene expression profile analysis of EBVaGC patients shows significant alterations in immune response genes, which probably are responsible for improved survival outcomes in patients [125]. Thirty-two (5.3%) EBVaGCs were identified, which have more CD8+ (*p* < 0.001) and Foxp3+ (*p* = 0.020) cell infiltration than EBV-negative GCs and have a higher 5-year overall survival (*p* = 0.003). Moreover, PD-L1 expression is linked with a poor 5-year OS (*p* = 0.002) [126]. EBVaGC is found to express high levels of PD-L1 in cancer and infiltrating immune cells. Tumor cells are producing PD-L1, which interacts with PD-1 on the surface of T-cells to escape from the immune system, while the high expression of PD-L1 on EBVaGC is correlated to tumor progression [127]. Moreover, the density of CD3+ T-lymphocytes (23.84 versus 12.76, *p* < 0.001) and CD68+ macrophages (9.73 ± 5.25 versus 5.44 ± 4.18, *p* < 0.001) are identified as significantly higher in EBVaGC compared with EBV-negative GC patients, while CD3+ T-cell density is associated with a higher 5-year OS of EBVaGC patients (*p* = 0.022) [128].

Furthermore, to detect the prognostic value of CD8+ cell infiltration and PD-L1 expression, Ma et al. classified 571 GC into four different TIMEs. The distribution of the four TIME types was 12.96% (type I, CD8+PD-L1+), 42.73% (type II, CD8−PD-L1−), 28.02% (type III, CD8−PD-L1+), and 16.28% (type IV, CD8+PD-L1−) in the complete cohort. Specifically, the proportion was 25.81% and 12.22% (type I), 19.35% and 44.07% (type II), 16.13% and 28.70% (type III), and 38.71% and 15.00% (type IV) in EBVaGC and EBV-negative GC, respectively, displaying a significant difference among them (*p* < 0.001), while the log-rank test revealed that type IV had the best 5-year OS (*p* < 0.001) [126].

In addition, Liu et al. showed the efficacy of immunotherapy in the treatment of EBVaGC, since 2000. This study included 300 GC patients (Asian) and the evaluation showed that 59.3% had PD-L1Cps ≥ 1 and 40.7% had PD-L1Cps < 1. PD-L1Cps ≥ 1 cases were significantly linked with high microsatellite instability (MSI-H) (*p* < 0.001), with stage I tumor (*p* = 0.022), positive EBV status (*p* = 0.008), and positive H. pylori status (*p* = 0.001) [129]. The above data implied that patients with EBV-positive GC could benefit from immunotherapy administration.

### 5.2. EBV-Associated Intrahepatic Cholangiocarcinoma (iCCA)

Intrahepatic cholangiocarcinoma (iCCA) is the second most frequent malignancy of the liver, while it presents a poor resectability, limited current therapeutic options, and low survival rates. The total data of EBV-associated ICC (EBVaICC) study are limited since it concerns a small portion of the total iCCA cases. Moreover, the study of Huang et al. revealed that this subtype was detected only in 6.6% (20/303) of the patients. Of note, EBV was not associated with perihilar cholangiocarcinoma (pCCA), combined mixed hepatocellular, cholangiocarcinoma (cHCC-CCA), distal cholangiocarcinoma (dCCA), as well as HCC. As in EBVaGC, EBVaICC has a prevalent histological appearance of the lymphoepithelioma-like subtype (LEL subtype) in a percentage of 45%, while the percentage of this histological type is only 0.7% in non-EBVaICC. EBVaICC (LEL subtype) was related to a significantly higher 2-year OS rate (89%) in comparison with conventional EBVaICC (36%) and non-EBVaICC (38%) (*p* = 0.028) [130]. Additionally, Chan et al. showed comparable results, with EBVaICC (LEL subtype) having a significantly better 2- and 5-year overall survival than various subtypes of ICC (100% versus 52.8%, and 100% versus 13.2%, respectively, *p* = 0.003) [131].

The CD3+ T-cells contained in EBVaICC were the most prevalent category of TILs (84.4%), while CD20+ B-cells and CD68+ TAMs had a percentage of 9.4% and 6.1%, respectively. CD8+ T-cells constituted the 71.4%, while FoxP3+ T-cells and CTLA-4+ T-cells accounted for the 15.0% and 13.6%, respectively. The proportion of CD20+ B-cell and CD8+ T-cell populations were significantly greater in EBVaICC versus non-EBVaICC cases, while the densities of TILs, (CD20+ B-cells, CD3+ T-cells, CD8+ T-cells, FoxP3+ T-cells, CTLA-4+ T-cells, CD68+ TAMs, HLA-DR+ M1 TAMs, and CD163+ M2 TAMs were also significantly higher in EBVaICC compared with non-EBVaICC [130].

PD-L1 expression in tumor cells, as well as PD-1 and PD-L1 expression in TILs were overexpressed in EBVaICC. A total of 95.0% (19/20) of EBVaICCs expressed PD-L1 in tumor cells (IRS score ≥ 3), but only the 22.3% (63/283) of non-EBVaICCs case expressed PD-L1 (*p* < 0.0001). Moreover, 100% (20/20) of EBVaICCs were positive for PD-L1 in TILs (IRS score ≥ 1), but only 56.5% (160/283) for non-EBVaICCs cases (*p* < 0.0001). In addition, a total of 95.0% (19/20) of EBVaICCs expressed PD-1 in TILs (IRS score ≥ 1) and only 64.0% (181/283) of non-EBVaICCs (*p* = 0.005). Furthermore, the sample of ICCs was categorized into a TMIT (type I, II, III, and IV) based on IHC results as follows: Type I, 53 samples (17.5%); Type II, 123 (40.6%); Type III, 29 (9.6%); and Type IV, 98 (32.3%). TMIT was significantly associated with overall survival in ICC (*p* = 0.014). The TMIT I subgroup had the best survival advantage, whereas the TMIT III subgroup had the poorest survival value. EBVaICC was significantly related to TMIT I since 90% (18/20) of EBVaICCs belonged to TMIT I, while the value was only 12.4% (35/283) for non-EBVaICCs (*p* < 0.0001) [130]. The aforementioned data of this study implied that patients with EBVaICC could be possible candidates for immunotherapy, with a better survival for those presenting the LEL subtype.

## 6. Human Papilloma Virus (HPV)

Human papilloma virus is a small DNA virus with a specific tropism for squamous epithelia, while they have been detected in 202 different HPV types. The HPV types that induce infection of the mucosa are additionally categorized into high- and low-risk groups, based on the possibility of developing malignancy in the tissue. More specifically, low-risk HPVs (HPV6 and HPV11) cause benign warts, but high-risk HPVs (HPV16 and HPV18) cause premalignant squamous intraepithelial neoplasia that can be subsequently developed into cancer [132]. In addition, HPV is related to numerous types of cancer, including cervical, vaginal, vulvar, anal, penile, and head-and-neck cancer. HPV-associated head-and-neck cancers, such as oropharyngeal squamous cell carcinoma (OPSCC) and oral squamous cell carcinomas (OSCC) are dramatically increased in recent years, mainly in men under 50 years old [133].

### 6.1. HPV-Associated Oral Squamous Cell Carcinomas (OSCC)

OSCC is the eighth most frequent cancer in males and the 14th in females in the US. Even with the current therapeutic modalities, OSCC causes approximately 700,000 new cases and around 380,000 deaths worldwide, while the 5-year survival remains only 64.4%. The prognosis of this type of cancer is poor since around 50% of patients have cervical lymphatic metastasis at the time of diagnosis. The incidence of high-risk HPV types in oral SCC is very low (<4%), with almost all HPV-positive cases being attributed to HPV16. The possibility of identifying HPV in normal oral mucosa is significantly smaller (10%) than in OSCC oral mucosa (46.5%) [134,135].

A recent report analyzed 160 HPV-negative OSCC lesions based on the expression of HLA-I antigens and the components of the antigen processing machinery (APM). Tumoral HLA-I APM component expression was further classified into the three main phenotypes, namely HLA-I^high^/APM^high^, HLA-I^low^/APM^low^, and HLA-I discordant^high/low^/APM^high^. In addition, the HLA-I^high^/APM^high^ group presented the highest incidence of intra-tumoral CD8+ T-cells and the lowest number of CD8+ T-cells, with the presence of FoxP3+ cells, while the patients of this group had the shortest survival despite a high intra-tumoral CD8+ rate [136].

In another study, patients with high FoxP3 expression presented a worse prognosis than patients with low expression. In the same study, the 5-year survival in HPV-positive and HPV-negative OSCC patients demonstrated a slight difference in the prognosis; however, there was no statistical importance, with a better prognosis identified in HPV-positive patients [137].

Furthermore, another study revealed a non-statistically significant correlation in survival between HPV-positive and HPV-negative OSCC, while no difference was found in the expression of PD-L1. From the above, it is concluded that immunotherapy possibly would not offer special OS of patients with OSCC, whether they are HPV-positive or HPV-negative [138].

### 6.2. HPV-Associated Anal Squamous Cell Carcinomas (ASCC)

Anal squamous cell carcinomas (ASCC) constitute a frequent malignancy located in the anal canal, which is lined by a mucosa-associated lymphoid tissue system (MALT) that is composed of several immune cells, including CD8+ and CD4+ T-cells, as well as Langerhans cells. It has been shown that 90% of the ASCC cases are induced by human papillomaviruses (HPV), which exhibit oncogenic properties [139].

HPV has a circular double-stranded DNA, which expresses several protein molecules, such as E7 and E6, which suppress the action of Retinoblastoma and TP53 proteins, which constitute pivotal tumor suppressor proteins. This phenomenon leads to the misappropriation of the cell cycle of the host. A high prevalence of HPV-related ASCC is reported in human immunodeficiency virus (HIV)-infected or iatrogenically immunosuppressed patients, as well as in men who have sex with men (MSM), while precursors of this malignancy are considered high-grade squamous intraepithelial lesions (HSILs), which can progress to ASCC, due to impaired immune responses in the anal mucosa even after treatment with electrocautery ablation [140,141,142].

A better understanding of the characteristics and the microenvironment of this malignancy is considered crucial for the discovery of further druggable targets. ASCC microenvironment is also composed of several cells, including TAMs, Tregs, as well as MDSCs, which can promote disease progression. First, CD8+ cytotoxic T-cells can be excluded from the ASCC microenvironment, which is mainly induced via MDSCs, that also promotes their apoptosis. Confirmed HPV-associated ASCCs are considered more immunogenic, expressing an enhanced peritumoral amount of TILs in the microenvironment, which is associated with a better prognosis, in comparison with non-HPV-related ASCCs, while the increased amount of TILs is related to a favorable response to chemoradiation [142,143].

Currently, immunotherapeutic strategies are promising for the management of this malignancy, combined with chemoradiation treatment (CRT). Radiation therapy induces the activation of dendritic cells, which leads to successful tumor recognition and elimination via CD8+ T-cells, whereas it also enhances the expression of PD-L1 on the surface of cancer cells, which increases the favorable effect of immune checkpoint blockades. Several clinical studies access the utilization of immune checkpoint blockade in ASCC, especially PD-1 and PD-L1 inhibitors for localized ASCC cases with or without micrometastases, which have been proved to be beneficial, especially in combination with radiotherapy. In malignancies that are characterized by an overexpression of PD-1/PD-L1 and CD8+ T-cells, such as ASCC, it has been demonstrated that the immune checkpoint blockade provides favorable effects [144]. Nivolumab (PD-1) has been assessed for advanced metastasis, which was previously unsuccessfully treated with ASCCs. Combinational treatment with nivolumab and ipilimumab (CTLA-4 inhibitor) is also accessed in NCT02408861, a phase I trial for HIV-related ASCCs that were also previously unsuccessfully treated with other agents [145,146,147]. Another therapeutic modality for HPV-related tumors is the ADXS11-001 vaccine that targets E7 on HPV cells (a phase I/II clinical study) [148,149]. Figure 2 presents the viruses associated with digestive system cancers.

## 7. Conclusions

Conclusively, viruses play an important role in carcinogenesis, both in the histological characteristics of each tumor and in TIME. TME components and characteristics have a key role in malignancy immunosurveillance, as well as can significantly modify the response to immunotherapeutic agents. At the level of post-operation prognosis, the ratio between CD8+ T-cells and Tregs is considered pivotal for the effective anti-cancer immune response. A good prognosis is demonstrated in the cases that present a high immune cell infiltration. However, overexpression of PD-1/PD-L1 immune checkpoints and interferon-related genes are closely associated with a worrisome prognosis. Infection with viruses creates higher immune cell infiltration and this is established when comparing tumors associated with a virus to tumors not associated with an infection (Figure 3). The application of immunotherapy could prove beneficial for optimal management. Multiple immunotherapeutic modalities can be used, such as vaccines against immunogens, the so-called tumor-associated antigens (TAAs), immune checkpoint blockade, as well as adoptive cell therapy. Some of the potential immunotherapeutic strategies are considered for the inhibition of the PD-1/PD-L1 axis, which is implied by the fact that PD-1/PD-L1 significantly modifies the T-cell response. Moreover, the utilization of CTLA-4 blockade is considered beneficial, due to the fact that it activates T-cells, by inhibiting the interaction between CD86, CD80, and CTLA-4. Consequently, the testing of the above modalities in clinical trials (both individually and in combination) may yield new therapeutic approaches for the benefit of specific patients. Finally, further research is required to investigate the correlation between virus infection and the immune microenvironment toward the optimal therapeutic management of the aforementioned malignancies.

## Figures and Tables

**Figure 1 ijms-23-13612-f001:**
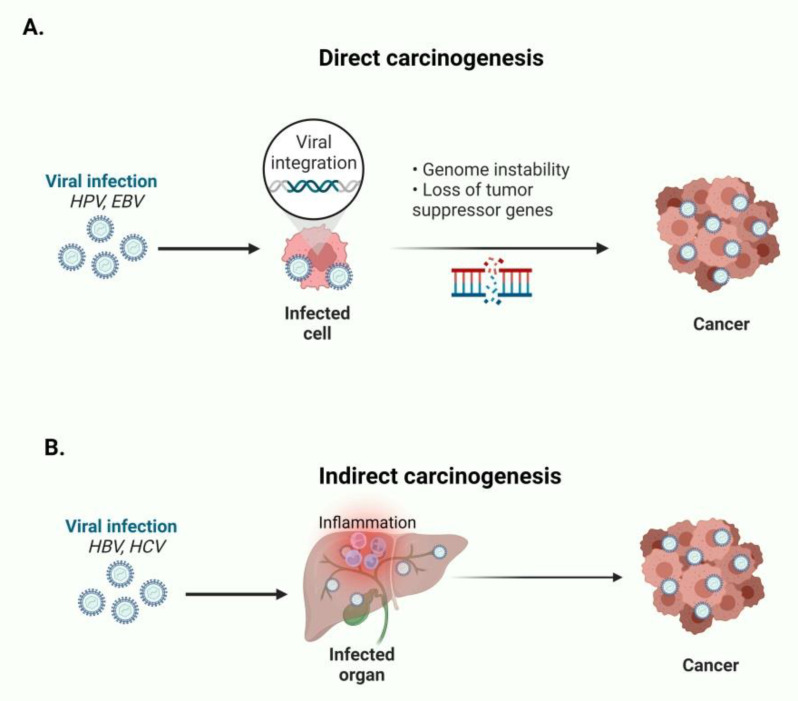
(**A**) HPV and EBV directly cause oncogenesis by expressing oncogenic proteins and establish genome instability. (**B**) HBV and HCV indirectly cause carcinogenesis by establishing chronic inflammation of the infected organs. This figure was created based on the tools provided by Biorender.com (accessed on 9 March 2022).

**Figure 2 ijms-23-13612-f002:**
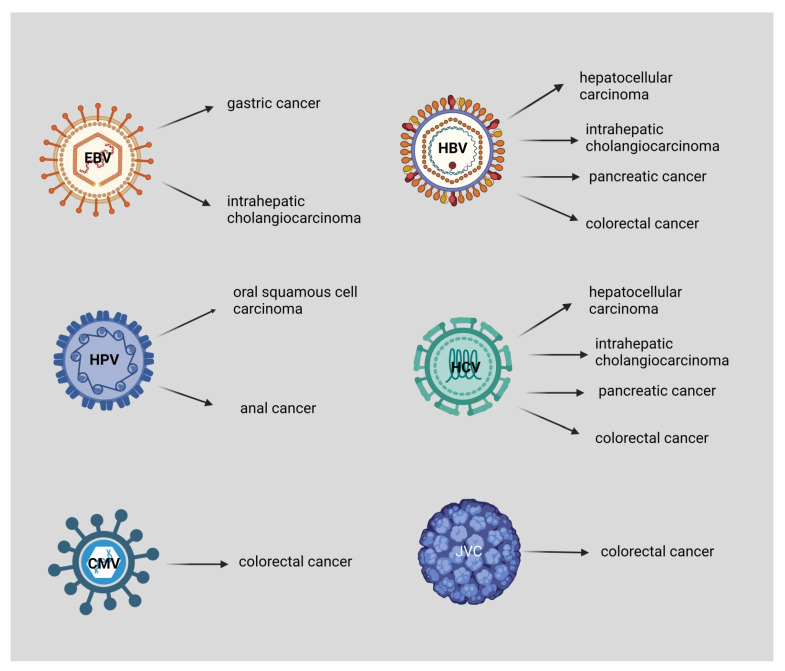
EBV is associated with gastric cancer and intrahepatic cholangiocarcinoma, HPV with oral squamous cell carcinoma, and anal cancer. Both HBV and HCV are associated with hepatocellular carcinoma, intrahepatic cholangiocarcinoma, pancreatic cancer, and colorectal cancer. Moreover, CMV and JCV are associated with colorectal cancer. This figure was created based on the tools provided by Biorender.com (accessed on 3 September 2022).

**Figure 3 ijms-23-13612-f003:**
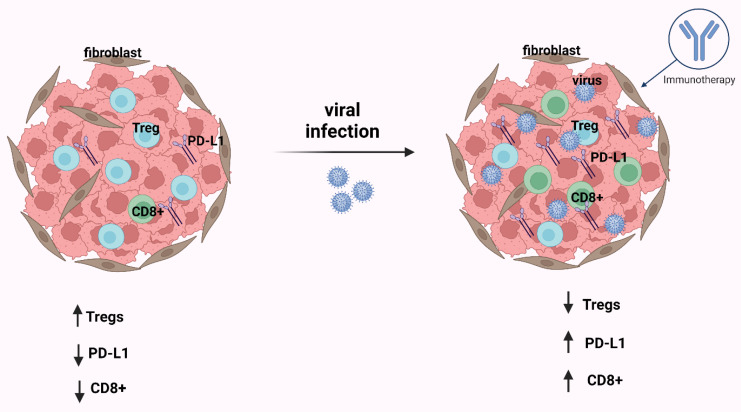
After infection by an oncogenic virus, in most cases in the tumor immune microenvironment (TIME), CD8+ T-cells and PD-L1 are increased and Tregs are decreased. The consequence of these events is that the immunotherapy is effective. This figure was created based on the tools provided by Biorender.com (accessed on 25 October 2022).

**Table 1 ijms-23-13612-t001:** Overview of HCV components that induce hepatic carcinogenesis.

HCV Non-Structural and Structural Proteins	Effect
Core protein, NS2, NS3, NS5A	Inhibition of apoptosis
Core, NS5A	Reactive oxygen species → TGF-β → HSCs activated → fibrogenesis
E1, E2, NS5A	Chronic inflammation via cytokine secretion
E2, core, NS2, NS3, NS5A	Cell proliferation
Core	EMT Mutagenesis and genome instability Angiogenesis via VEGF Lipid metabolism modification → steatosis

## Data Availability

Not applicable.

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
