# Peer review of "Immune Microenvironment and Immunotherapeutic Management in Virus-Associated Digestive System Tumors"

_ijms, 2022, doi:10.3390/ijms232113612_

Round 1
Reviewer 1 Report
This review presented the association between the TIME and HBV, HCV, EBV, as well as HPV associated digestive malignant tumor. These content seems informative and interesting, however, there have several issues affecting the manuscript suitable for publication.
1. The title should use the words "digestive tumors" rather than "gastrointestinal tumors" according to the contents below.
2.The authors mentioned four types of immune responses in abstract and introduction, but never saw it again in the following manuscript. More interpretation should be presented.
3. As a review, more details about Immune microenvironment and clinical implications should be added in every subheading paragraph.
Author Response
October 27, 2022
RE: REVISED REVIEW Article (ijms-1959971 REVISED)
We would like to thank the Reviewer for his/her thoughtful evaluation of our manuscript and for his/her most welcome comments/suggestions. Accordingly, we have now revised our manuscript thoroughly to reflect these comments.
In the revised Text all changes/additions/modifications made in response to the Reviewer’s points (including linguistic revisions throughout the manuscript) are marked up in red.
Please find below a point-by-point response to the issues raised by Reviewer 1:
Reviewer 1
This review presented the association between the TIME and HBV, HCV, EBV, as well as HPV associated digestive malignant tumor. These content seems informative and interesting, however, there have several issues affecting the manuscript suitable for publication.
Response: We thank the Reviewer for his/her evaluation of our work.
- The title should use the words "digestive tumors" rather than "gastrointestinal tumors" according to the contents below.
Response: Following the Reviewer’s suggestion we changed the term “gastrointestinal tumors” to “digestive system tumors” in the title and wherever needed in the manuscript.
- The authors mentioned four types of immune responses in abstract and introduction, but never saw it again in the following manuscript. More interpretation should be presented.
Response: This categorization is carried out only in EBV, so the relevant reference to the four types of immune responses was transferred there.
- As a review, more details about Immune microenvironment and clinical implications should be added in every subheading paragraph.
Response: We added information about the tumor immune microenvironment in sections where we found relevant studies. In some, unfortunately, there are no studies on the tumor immune microenvironment, so we have added the necessary comment. We should note here that we were in a dilemma from the beginning about whether we should include these sections in the review since we refer to the immune microenvironment. We finally decided to include them because primarily we wanted to highlight the viruses-cancer correlation. By the term clinical implications, we meant the therapeutic approaches that may exist, especially for using immunotherapy. We realize that the term is not appropriate and may create confusion, therefore we have changed it to immunotherapeutic management (see also change in title).
Note: In addition to the above, an extensive review on English spelling and an improvement in the use of English expressions and abbreviation definition has been performed in the revised manuscript.
Trusting that we have adequately addressed the Reviewer’s concerns, we would like to thank him/her for his/her help in improving significantly our work.
With kind regards,
Prof. Athanasios G. Papavassiliou, MD, PhD
Corresponding author

Reviewer 2 Report
The manuscript entitled “Immune Microenvironment and Clinical Implications in Virus- 2 Associated Gastrointestinal Tumors” by Saranatis et al. seems interesting. Although authors have written a very informative and have tried to describe the role of many of the viruses such HBV, HCV, EBV, and HPV in the development and progression of Gastrointestinal Tumors with emphasizing the role of altered immune microenvironment but still there is a much scope to improve it, which will help in understanding the relationship between the immune microenvironment and its therapeutic relevance in virus-associated GI cancer. I have a few suggestions for authors, which are mentioned below:
1. Authors should write a paragraph on each section about the implication of two crucial tumor-associated immune cells, namely macrophages and T regulatory cells in oncogenic virus-mediated gastrointenstinal tumors.
2. Relevant references are missing (such as lines 128-132) at several places, so authors should read the manuscript carefully and cite the proper references wherever needed.
3. Reference and content are mismatched (lines 193-200) references 36 and 37, please check and do the needful modification.
4. Authors should discuss the clinical implications of the immune microenvironment modulation for gastrointenstinal tumors in each section as it is one of the major components of this manuscript.
5. Several viruses like JC virus is also well associated with GI cancer is not discussed; it will be good if they include such viruses as well.
6. More pictorial or graphical representations are needed for the better understanding.
7. Throughout the manuscript, authors have majorly cited review articles, so, they should cite relevant research articles at most of the places rather than review articles.
Author Response
October 27, 2022
RE: REVISED REVIEW Article (ijms-1959971 REVISED)
We would like to thank the Reviewer for his/her thoughtful evaluation of our manuscript and for his/her most welcome comments/suggestions. Accordingly, we have now revised our manuscript thoroughly to reflect these comments.
In the revised Text all changes/additions/modifications made in response to the Reviewer’s points (including linguistic revisions throughout the manuscript) are marked up in red.
Please find below a point-by-point response to the issues raised by Reviewer 2:
Reviewer 2
The manuscript entitled “Immune Microenvironment and Clinical Implications in Virus- 2 Associated Gastrointestinal Tumors” by Saranatis et al. seems interesting. Although authors have written a very informative and have tried to describe the role of many of the viruses such HBV, HCV, EBV, and HPV in the development and progression of Gastrointestinal Tumors with emphasizing the role of altered immune microenvironment but still there is a much scope to improve it, which will help in understanding the relationship between the immune microenvironment and its therapeutic relevance in virus-associated GI cancer. I have a few suggestions for authors, which are mentioned below:
Response: We would like to thank the reviewer for the apt comments and constructive remarks.
- Authors should write a paragraph on each section about the implication of two crucial tumor-associated immune cells, namely macrophages and T regulatory cells in oncogenic virus-mediated gastrointenstinal tumors.
Response: We added information about the tumor immune microenvironment in sections where we found relevant studies. In some, unfortunately, there are no studies on the tumor immune microenvironment, so we have added the necessary comment. We should note here that we were in a dilemma from the beginning about whether we should include these sections in the review since we refer to the immune microenvironment. We finally decided to include them because primarily we wanted to highlight the viruses-cancer correlation.
- Relevant references are missing (such as lines 128-132) at several places, so authors should read the manuscript carefully and cite the proper references wherever needed.
Response: We have added relevant references at the text place you mention as well as in the entire manuscript. We trust this issue has now been corrected.
- Reference and content are mismatched (lines 193-200) references 36 and 37, please check and do the needful modification.
Response: We thank the Reviewer for pointing this out. Accordingly, we have made the appropriate modifications.
- Authors should discuss the clinical implications of the immune microenvironment modulation for gastrointenstinal tumors in each section as it is one of the major components of this manuscript.
Response: By the term clinical implications, we meant the therapeutic approaches that may exist, especially for using immunotherapy. We realize that the term is not appropriate and may create confusion, therefore we have changed it to immunotherapeutic management (see also change in title).
- Several viruses like JC virus is also well associated with GI cancer is not discussed; it will be good if they include such viruses as well.
Response: Following the Reviewer’s suggestion we added information about JC and CMV-related CRC. However, data about tumor microenvironment and JC virus are not sufficient in the literature.
- More pictorial or graphical representations are needed for the better understanding.
Response: We agree with the Reviewer’s recommendation and we added a new figure, Figure 3, for a better understanding of the manuscript.
- Throughout the manuscript, authors have majorly cited review articles, so, they should cite relevant research articles at most of the places rather than review articles.
Response: Following the Reviewer’s suggestion we included relevant research articles at most of the places, especially those referring to clinical studies.
Note: In addition to the above, an extensive review on English spelling and an improvement in the use of English expressions and abbreviation definition has been performed in the revised manuscript.
Trusting that we have adequately addressed the Reviewer’s concerns, we would like to thank him/her for his/her help in improving significantly our work.
With kind regards,
Prof. Athanasios G. Papavassiliou, MD, PhD
Corresponding author

Round 2
Reviewer 1 Report
This revised manuscript has addressed most of my concerns raised based on the previous version, and has significantly improved its quality. Still, several suggestions are proposed here to further improve the clarity of this manuscript.
1. In line 171, “An increased risk of gastric cancer is also reported in patients with HBV infection, as was demonstrated in meta-analytic studies [55]”, what does this have relationship with CRC?
2. In line 185-186, “A study by Cheng et al. revealed that efficacy and OS were similar between patients with HBV infection and non-HBV patients receiving anti-PD-1 therapy. Therefore, CRC-HBV-positive patients can benefit from immunotherapy”, which seems contradictory.
Author Response
Reviewer 1 (Round 2)
This revised manuscript has addressed most of my concerns raised based on the previous version, and has significantly improved its quality. Still, several suggestions are proposed here to further improve the clarity of this manuscript.
Response: We thank the Reviewer for his/her evaluation of our revised manuscript.
- In line 171, “An increased risk of gastric cancer is also reported in patients with HBV infection, as was demonstrated in meta-analytic studies [55]”, what does this have relationship with CRC?
Response: We thank the Reviewer for the remark, the sentence was removed from the manuscript
- In line 185-186, “A study by Cheng et al. revealed that efficacy and OS were similar between patients with HBV infection and non-HBV patients receiving anti-PD-1 therapy. Therefore, CRC-HBV-positive patients can benefit from immunotherapy”, which seems contradictory.
Response: Τhere are positive results from immunotherapy in colorectal cancer. therefore, the same effectiveness is a positive indication for CRC-HBV-positive patients as well
Note: In addition to the above, an extensive review on English spelling and an improvement in the use of English expressions and abbreviation definition has been performed in the re-revised manuscript.
Trusting that we have adequately addressed the Reviewer’s additional concerns, we would like to thank you for your help in improving further our work.
With kind regards,
Prof. Athanasios G. Papavassiliou, MD, PhD
Corresponding author
IJMS Editorial Board member
